# An Integrated Model of Destination Attractiveness and Tourists’ Environmentally Responsible Behavior: The Mediating Effect of Place Attachment

**DOI:** 10.3390/bs13030264

**Published:** 2023-03-16

**Authors:** Ting Li, Chenmei Liao, Rob Law, Mu Zhang

**Affiliations:** 1Shenzhen Tourism College, Jinan University, Shenzhen 518053, China; 2Asia-Pacific Academy of Economics and Management, Department of Integrated Resort and Tourism Management, Faculty of Business Administration, University of Macau, Macau 999078, China

**Keywords:** destination attractiveness, place attachment, tourists’ environmentally responsible behaviors, ecological protection

## Abstract

This study aimed to empirically explore the relationship between destination attractiveness and tourists’ environmentally responsible behavior (TERB), as based on self-regulated attitude theory. In this paper, we have divided destination attractiveness into two aspects: the attractiveness of a facility’s services, and that of the sightseeing experience, so as to build a structural equation model with mediation analysis. For our research, we selected Wolong National Nature Reserve in Sichuan as the site of our case study, and we conducted a survey using a questionnaire. We then analyzed the path using a structural equation model. Our results show: (1) two elements of destination attractiveness have significantly positive effects on TERB; and (2) place attachment exerts a mediating effect among the attractiveness of the facility’s services, that of the sightseeing experience, and TERB. Therefore, enhancing destination attractiveness and tourists’ emotional attachments to locations could help to promote the implementation of TERB and the achievement of sustainable tourism development.

## 1. Introduction

In recent years, the rapidly developing tourism industry has brought considerable economic benefits to tourism locations, but it has also destroyed the ecological environment of some of them. In addition, environmental problems disrupt the development of tourism, bringing unprecedented pressure on the sustainable development of tourism destinations [1]. Therefore, the sustainable development of tourism has received considerable worldwide attention [2]. Tourism destinations are the foundation of sustainable tourism development, and as a consequence of the popularization and normalization of tourism activities in China, negative environmental impacts have occurred in most of the destinations, and tremendous negative eco-environmental consequences have occurred due to tourists’ behaviors [3,4]. Tourists are critical stakeholders in the sustainable development and environmental management of tourism destinations [5]. Relevant research has shown that guiding and encouraging mass tourists to take the initiative in implementing environmental protection behaviors is an effective means of dealing with ecological and environmental problems in tourism destinations [6].

TERB is considered to be the behavior of tourists during travel that contributes to environmental protection and the well-being of the destination. It requires tourists to have a strong sense of responsibility for local natural and human environmental factors [4,7]. Especially in nature-based destinations, TERB significantly influences the destination’s environment, which determines the maintenance and sustainability of the destination’s attractiveness [8]. Therefore, TERB is an essential indicator of sustainable tourism, and increasing TERB can enhance the sustainability of destinations [5]. Knowing the factors affecting tourists’ environmentally responsible behavior (TERB) and effectively guiding its spontaneous implementation is an urgent step to protect the ecological environment.

Destination attractiveness is an important factor that influences travelers’ decisions. Harmful, negative environmental consequences of tourism activities in natural environments may make destinations lose their attractiveness and sustainability [3]. However, the attractive features of the destination can stimulate tourists’ awareness of environmental issues. When tourists see a spectacular destination environment, they will pay more attention to environmental issues and put TERB into practice [9]. Regarding the effect of destination attractiveness, most scholars prefer to study the effect on individual consumption attitudes, such as tourist loyalty, tourist revisit intention, place attachment, and their willingness to pay for tourism eco-compensation; however, few have addressed the effect on individual behaviors [10,11,12,13,14,15,16]. A significantly positive effect of attractiveness on place attachment and willingness to pay has been explored, and during the examination of the sense of place, tourism attractiveness has been found to have a significantly positive effect on the formation of a sense of place; however, few studies have deeply explored the mechanism of the influence of destination attractiveness on individual behavioral responses. By investigating tourists in the Penghu Islands, some researchers found that, the stronger the destination attractiveness as perceived by tourists, the more TERB will be implemented [17]. In addition, other scholars found that perceived levels of tourism resource quality, facilities, and services, as reflected in tourism imagery, all have significantly positive effects on TERB; however, the study neglected the effects of changes in tourists’ emotions on their behaviors as a result of the reactions generated during their interactions with a destination, such that the formation mechanism underlying TERB could not be further explored [18,19]. Although relevant research results on TERB are relatively fruitful—they have been conducted on bird-watching tourists, residents, and ecotourism-site tourists, the number of relevant studies targeting tourists in national nature reserves is relatively small. Therefore, investigating the relationship between destination attractiveness and TERB is of strong theoretical value [6,20,21].

Place attachment describes the emotional attachment between a person and a place, and it is often involved in the study of tourist behaviors [22]. Place attachment is divided into four dimensions, namely, place identity, place dependence, place influence, and place social connection, so as to better explore its influence on TERB [23]. Previous research on place attachment has found that TERB in tourism contexts is significantly and positively influenced by place attachment, verifying that a deeper sense of place attachment will promote the implementation of TERB [24]. Destination attractiveness reflects the perception of visitors to the destination; it depends on whether the destination meets the needs of tourists and how tourist perceptions will influence the formation of place attachment [9]. Unfortunately, researchers have not explored how destination attractiveness plays a role in human–place interactions or how it further influences the implementation of TERB. Therefore, investigating whether and how place attachment plays a mediating role in the relationship between destination attractiveness and TERB is worthwhile. In previous studies on affective attachment in human–place interactions, scholars have usually linked human–place affective factors to tourism marketing, tourism experience, and tourist loyalty, whereas few studies have explored the implications for ecological protection [25,26]. Research on the mechanism of destination attractiveness and place attachment on TERB has not received enough attention. This paper investigates sustainable behavior from a place theory perspective, empirically analyzes the interaction between destination attractiveness and TERB, and attempts to examine the mediating role of place attachment. The results of this study have significance for guiding the environmental protection of tourism destinations and adjusting the psychological perceptions of tourists to achieve sustainable tourism development.

There exists a lack regarding theories about the causal relationships among destination attractiveness, place attachment, and TERB. Although cultivating TERB has been emphasized, the effect of a destination’s unique fascination on TERB unknown [27]. TERB is an individual behavior, and destination attractiveness is a cognitive component of the “cognitive–emotional–intentional” response, which is a key pre-variable of individual behavioral responses. In order to support this viewpoint in the present study, we chose self-regulatory attitude theory as a theoretical support. Furthermore, open protected areas are places that can provide the products of natural experiences and education for visitors to stimulate environmentally responsible behavior, leading to the achievement of sustainability [28]. Therefore, as our case study site, we chose Wolong Nature Reserve in Sichuan, which is the largest nature reserve in Sichuan Province, with the most complex natural conditions and the rarest flora and fauna.

Accordingly, the present study contributes to the gap in verifying the relationships among destination attractiveness, place attachment, and TERB. The specific research objectives of this study were: (1) to apply self-regulated attitude theory to empirically study the relationship between destination attraction and TERB; (2) to test the mediating role of place attachment and investigate how place attachment plays a mediating role in the relationship between destination attractiveness and TERB.

## 2. Literature Review and Hypothesis

### 2.1. Theoretical Foundation

Self-regulatory attitude theory was developed by Bagozzi based on the cognitive evaluation theory of emotions, which extends the “evaluation → emotional response → behavioral intention” self-regulatory process [29]. This theory states that individuals often evaluate their past, present, or future situations, and as a result of this evaluation, they develop corresponding emotions that affect their behavior, finally presenting a continuous process of “cognitive → affective → behavior” [30]. Studying the cognitive process of individuals includes the measurement of the difference between the actual perceived outcome and the individuals’ expectations. When individuals experience something that meets or exceeds their expectations, “result–expectation” congruence is achieved, and a positive affective response is triggered, leading to positive behavioral intentions. However, when individuals have a poor experience or have experienced something that falls far short of their expectations, this situation leads to a contradiction between “outcome and expectation”, stimulating negative emotions and leading to negative behavioral intentions. When certain conditions are met, the intention is transformed into a behavioral response [31].

### 2.2. Relationship between Destination Attractiveness and TERB

Many studies have confirmed that destination attractiveness is an important factor that influences travelers’ decisions. Early research focused on the conceptual definition of destination attractiveness, mainly based on two perspectives: supply and demand. From the supply perspective, the quantity and quality of destination tourism attractions are the measurement dimensions of destination attractiveness. On the other hand, the demand perspective is based on the tourists’ perceptions and interests regarding a destination’s attributes. In this paper, to be consistent with the previous literature, destination attractiveness is defined for the purpose of studying tourist behavior as an aggregate that is composed of each attribute of the destination that attracts tourists, and it is a comprehensive perception and assessment of the overall attractiveness of the destination by tourists [31,32].

Scholars have paid more attention to the influencing factors of destination attractiveness. Formica and Sandro used supply and demand indicators to study the influencing factors of destination attractiveness, and they examined the influence of cognitive and affective imagery on destination attractiveness [32,33,34]. Reitsamer and Brunner-Sperdin argued that the components of destination attractiveness are mainly destination accessibility, infrastructure, scenic beauty, and local community, which are essential for destination image formation and the memory of the stages of the tourists’ destination experience [11]. 

Moreover, different approaches have been adopted to measure destination attractiveness, including multi-attribute and single-item measures. Tourism destinations have various attributes that are different from tourist activities, such as transportation conditions, history and culture, and accommodation facilities, which are the attributes that make destinations attractive to tourists, drawing them away from their permanent residences [24,25,26]. Some scholars have suggested that tourism destination attractiveness is highly correlated with the level of tourism attraction and the quantity and quality of tourism facilities and service [23]. Formica and Sandro identified four dimensions of destination attractiveness: tourism services and facilities, cultural history, rural accommodations, and outdoor recreation, representing the portfolio of destination attractiveness. Laws grouped destination attributes into two main categories: innate characteristics, such as natural resources, and characteristics introduced mainly for tourists, such as accommodation and tourist activities [35]. Das, Mohapatra, Sharma, and Sarkar claimed that destination attractiveness might be measured in terms of natural attractions and facilities [36]. Following the studies of Laws and Das et al., we divided destination attractiveness into two dimensions: facility service attractiveness and sightseeing experience attractiveness.

Research on destination attractiveness often regards it as a destination attribute variable; the more attractive a destination is, the deeper the perception of the destination will be, which has a significant effect on the attitudes of tourists [11,37]. A high level of destination attractiveness provides a guarantee of satisfying the diverse functional needs of tourists, so destination attractiveness is a necessary condition for the formation of place attachment emotions [38]. Henkel argued that the attractiveness of tourist attractions has a considerably important influence on tourists’ behaviors in destination tourism, and Nadzirah confirmed that tourists who are satisfied with the attractiveness of a marine park tend to commit TERB [39,40]. However, the exact process of the effect of destination attractiveness on TERB has not yet been examined. This study investigated the mechanism of the influence of destination attractiveness on TERB by measuring the overall magnitude of destination attractiveness. Therefore, this study presents the following hypotheses:

**Hypothesis 1a** **(H1a).**
*Facility service attractiveness positively affects the implementation of TERB.*


**Hypothesis 1b** **(H1b).**
*Sightseeing experience attractiveness positively affects the implementation of TERB.*


In the research field of tourism geography, TERB has become a topic of interest and a frontier in the study of human–place relationships. Environmentally responsible behavior includes several concepts, such as environmental responsibility, environmental behavior, and environmentally friendly behavior. Lee and Jan defined TERB as a series of behaviors in which tourists can promote the environmental protection of tourism destinations, minimize local environmental damage, and avoid the disturbance of ecosystems [41]. Most scholars have shifted their attention from rational to emotional research on TERB. By strengthening the emotional management of TERB, TERB has become an important target behavior in promoting the realization of the sustainable development of an ecological environment in tourist destinations [42]. Cheung and Ma studied the TERB of dolphin-watching tours and found that motivation, interpretative knowledge enhancement, and satisfaction are all positively and significantly correlated with TERB [43]. Su and Swanson utilized a stimulus–organism–response framework to present and examine an integrated model that investigates consumption emotions (positive and negative) and tourist destination identification as mediating variables between perceived destination social responsibility and Chinese TERB [44]. Affective factors influence tourists’ willingness to behave in an environmentally responsible manner, and the emotional connection between tourists and the destination, along with the psychological identification that tourists develop with the destination, drive the formation of behavioral attitudes and behavioral intentions related to the environment [45]. Tourists’ perceived emotions have significantly positive effects on their willingness to pay for rare species conservation [46]. If TERB can be managed emotionally, such that tourists actively engage in TERB, then such behaviors can be more sustainable, which can improve tourists’ behaviors and reduce environmental destruction, as well as promoting green tourism. From the perspective of the human–place relationship, specific tourist destination traits are classified, and the influence mechanism on tourists’ place attachment and TERB can then be explored [47]. Moreover, emotional imagery makes a difference in place identity and place dependence, which have different degrees of influence on TERB [48].

Some scholars have divided TERB into two dimensions: self-restraint behaviors and conservation promotion behaviors, focusing on exploring the relationships among local knowledge needs, place attachment, and the TERB of national park visitors [49]. Others have examined TERB in urban nature settings, concluding that the place attachment of visitors positively contributes to self-restraint behaviors in TERB [50].

### 2.3. Mediating Role of Place Attachment

Place attachment is a widely accepted emotional connection that arises through the interaction of people and places, and it is considered a positive emotional connection between tourists and scenic spots [51]. Place attachment is widely recognized by scholars in the study of tourist behavior as a key factor that regulates tourists’ perceptions. Some scholars have considered place attachment as an antecedent variable of TERB, revealing that the emotional bonds tourists form with destinations have significant effects on their behaviors; other scholars have used place attachment as a mediating variable or as an additional driver, so as to study the relationships among satisfaction, local knowledge needs, tourist place imagery, value orientation, and TERB [17,23,52,53,54,55]. Destination attractiveness affects place attachment, resulting in a sense of responsibility, which, in turn, affects the implementation of environmental responsibility. However, the manner in which place attachment plays a role between destination attractiveness and tourist behavior is a question that must be explored. As the place attachment of tourists decreases, their intentions to implement TERB will also decrease. In summary, this paper proposes the following additional hypotheses:

**Hypothesis 2** **(H2).**
*Place attachment has a mediating effect between destination attractiveness and TERB.*


**Hypothesis 2a** **(H2a).**
*Facility service attractiveness positively affects place attachment.*


**Hypothesis 2b** **(H2b).**
*Sightseeing experience attractiveness positively affects place attachment.*


**Hypothesis 2c** **(H2c).**
*Place attachment can positively affect TERB.*


Figure 1 summarizes the research hypotheses formulated in this paper.

## 3. Methodology

### 3.1. Study Area

Wolong National Nature Reserve (102°51′~103°24′ E, 30°45′~31°25′ N) is located in Wenchuan County, Sichuan Province, China, and it covers an area of approximately 200,000 ha (see Figure 2) [56]. The selection of the case study site was based on the following reasons: (1) Wolong National Nature Reserve is the core area of the giant panda national park system pilot program, and the national parks are a platform for promoting environmental protection; (2) Wolong National Nature Reserve in Sichuan is one of China’s first comprehensive national reserves. There are more than 100 giant pandas distributed throughout the region, accounting for about 10% of the total number of giant pandas in China. Wolong Nature Reserve is actively engaged in conservation, scientific research, and community building, so that wildlife resources and alpine ecosystems, mainly those of giant pandas, are effectively protected. One of the important purposes of opening protected areas to the public is to provide tourists with natural experience products and education in order to stimulate their environmentally responsible behaviors and achieve sustainability; (3) Wolong National Nature Reserve in Sichuan had received 1.05 million visitors as of 9 December 2022, which shows a large market presence and a wide range of visitor sources. Therefore, the properly guided, environmentally responsible behavior of tourists can accelerate the green development of ecologically functional areas, and this will set a good example for other ecologically oriented tourism destinations.

### 3.2. Questionnaire and Measurement Scale Design

The questionnaire was developed in two stages. in the first stage, the questionnaire was designed on the basis of relevant literature, and it was divided into four sections: the demographic characteristics of tourists; destination attractiveness, which is further divided into facility service attractiveness and sightseeing experience attractiveness; place attachment; and TERB. For more details, see Table 1.
(1)The experimental scale of destination attractiveness was mainly based on the research results of Hu and Liu [31,57].(2)The place attachment scale drew on the studies of Zhou and Scannell [54,58].(3)The TERB scale was based on the research results of He and Fan [5,59].

**Table 1 behavsci-13-00264-t001:** Measurement elements.

Construct	Item	Source
Facility service attractiveness	A1: This destination has good accommodations. A2: This destination has easy access to transportation nearby. A3: This destination provides me with a lot of leisure fun.	Hu (1993) and Liu (2017) [31,57]
Sightseeing experience attractiveness	B1: This destination has beautiful scenery. B2: This destination has profound cultural heritage.	
Place attachment	C1: This destination is a special place for me. C2: The experience of traveling to this destination is unique. C3: This destination gives me more satisfaction than other national parks. C4: I like this destination more than other parks.	Zhou (2014) and Scanell (2010) [54,58]
TERB	E1: I will obey the policy regarding not entering the national park protection area. E2: I will convince my peers to adopt environmental protection behaviors. E3: I will avoid disturbing the flora and fauna in the scenic area during the tour. E4: I will voluntarily reduce or stop the corresponding activities if the scenic spot needs to be restored.	He (2018) and Wu (2022) [5,59]

In the second stage, after preparing the first draft of the questionnaire, relevant experts were invited to examine the applicability and accuracy of the questionnaire items. Following their comments, we revised the questionnaire items. All scales in the questionnaire used a five-point Likert scale, ranging from 1 (strongly disagree) to 5 (strongly agree).

### 3.3. Data Collection

In view of the research purpose, the experimental period was from January to March 2021. Considering the insufficient number of validly completed questionnaires, we conducted a supplementary survey in 2023. As of 5 February 2022, a total of 456 questionnaires had been collected. After excluding those questionnaires that were completed in less than 60 s, and were thus obviously invalid, a total of 375 valid questionnaires was retained for study, yielding an effective rate of 82.23%. The Kolmogorov–Smirnov test was used, with a *p* value = 0.149 > 0.05, indicating that the data obeyed a normal distribution pattern. We collected the questionnaires through convenience sampling, and the questionnaires were completed anonymously so as to reduce CMV at the source to protect privacy. Harman’s single-factor test was used to evaluate CMV, and the results show that the first factor accounted for 22.34% of the total variance. If it had been less than 40% of the total variance, we could have considered that no serious common method bias had occurred [60]. The collected data were then analyzed using SPSS 24.0 and Amos 24.0. The statistical analysis of the demographic characteristics of the respondents was conducted using SPSS 26.0 descriptive statistics to analyze the valid questionnaires (n = 375). About 45.6% of the respondents were male, and 54.4% were female; in terms of age, the group aged 18–30 years accounted for the major proportion; in terms of education level, 7.4% of the respondents had finished high school and junior high school, whereas 21.1%, 56%, and 15.5% had a bachelor’s degree, a master’s degree, or above, respectively, indicating that the respondents were capable of understanding the content of the questionnaire. The overall male-to-female ratio of the sample in this study was 1:1, and the surveyed population represented a variety of educational levels and age groups. Thus, the selected sample was fairly representative and suitable.

## 4. Data Analysis

### 4.1. Exploratory Factor Analysis

This study used SPSS 26.0 software to analyze the data afforded by the completed questionnaires. The scale was initially subjected to exploratory factor analysis, the results of which showed that the scale ultimately retained 21 measurement items (KMO = 0.912, *p* = 0.000 < 0.05). The data results showed that dividing the scale into four factors was reasonable. The cumulative variance contribution rate reached 75.318%, which exceeded the 60% extraction standard [61].

### 4.2. Reliability Test

Reliability tests examine the degree of internal consistency among the observed variables of each potential variable. A combination reliability of <0.60 was adopted, indicating the potential variable. The combination reliability was satisfactory. The internal consistency coefficient value (Cronbach’s α) of the overall scale of the study was 0.916, and the Cronbach’s α of facility service attractiveness, sightseeing experience attractiveness, place attachment, and TERB were 0.815, 0.746, 0.887, and 0.814, respectively, indicating that the scale of the study and each dimension met the test criteria for combined reliability.

### 4.3. Test Validity

In this paper, the validity of the questionnaire data was tested using Amos software. Table 2 shows that all the indicators of the questionnaire data (i.e., absolute fitting index, relative fitting index, and reduced fitting index) reached the ideal level, so the measurement model of this study has good fitting validity.

The convergent validity of the questionnaire was tested using the average variance extracted (AVE) value. According to the results (Table 3), the standardized factor load of each item was greater than 0.5, and the range was maintained between 0.688 and 0.845. The CR values of facility service attractiveness, sightseeing experience attractiveness, place attachment, and TERB were 0.74, 0.69, 0.87, and 0.85, respectively. The combined reliability of the four variables was greater than 0.7. The AVE values ranged between 0.50 and 0.63, which exceeded the recommended value of 0.5 [62], thereby confirming an excellent level of convergent validity of the model.

## 5. Hypothesis Testing

This study used a structural equation model to test the proposed hypotheses. Facility service attractiveness and sightseeing experience attractiveness were taken as the independent variables, and TERB was the dependent variable when we tested whether place attachment mediates the variables. A structural equation model was constructed, as shown in Figure 1. The path coefficient was calculated using the maximum likelihood estimation method, and the hypotheses were examined for the purpose of their verification. The results are shown in Table 4.

The hypotheses proposed in this paper all received empirical support from the data. Specifically, for H1a, the standardized path influence coefficient of facility service attractiveness on TERB was 0.835 (*p* < 0.001). For H1b, the standardized path influence coefficient of sightseeing experience attractiveness on TERB was 0.792 (*p* < 0.001). Therefore, H1a and H1b were verified, and the verification results were positively related to destination attractiveness and TERB, that is, the higher the degree of destination attractiveness, the higher the intention of TERB will be. H2a and H2b indicated the relationship between the two dimensions of destination attractiveness and place attachment. The results show a positive correlation between them. H3b indicated the relationship between place attachment and TERB. The results showed that tourists’ sense of control positively affects TERB (standardized path coefficient = 0.822, *t*-value = 4.674, *p* < 0.001). That is, during the tourism process, the stronger the place attachment of a tourist is, the easier the implementation of TERB will be.

## 6. Mediating Effect Test

On the basis of the determined paths, this study used a bootstrap method for the calculation, with 5000 iterations. As shown in Table 5, the coefficients were all significant (*p* < 0.001).

The first path is “Facility service attraction → Place attachment → TERB”. The indirect effect was 0.072, with a percentile inspection of 95%, and the interval was [0.064, 0.116]. The second path is “Sightseeing experience attractiveness → Place attachment → TERB”. The indirect effect was 0.081, with a percentile inspection of 95%, and the interval was [0.053, 0.097]. Neither of their intervals included 0. Therefore, a mediating effect was observed. In summary, place attachment mediates the relationship between destination attractiveness and TERB.

## 7. Discussion and Conclusions

### 7.1. Discussion

TERB is important for the protection of natural resources and the environment, as well as for ensuring the sustainability of nature reserves. Therefore, activating TERB is crucial to mitigating the negative impact of tourism on the natural environment. However, the causes of TERB are complex and there are many factors that influence TERB [63].In this study, on the basis of self-regulated attitude theory, destination attractiveness was selected as the variable of tourists’ evaluation of a destination, and it was measured using the two elements of facility service attractiveness and sightseeing experience attractiveness. In addition, place attachment was introduced as the variable for tourists’ emotional responses, and the response behavior of tourists was reflected by TERB. A model of the mechanism of the influence of destination attractiveness on TERB was drawn and empirically tested using tourists who visited Sichuan’s Wolong Nature Reserve as the respondents.

The empirical test results further reveal the effect of the emotion-driven path of destination attractiveness on TERB. The results indicate that destination attractiveness, as an external stimulus, can affect tourists’ affective attitudes, and when tourists develop high levels of place attachment, they are more inclined to implement TERB. This finding suggests that local management should maintain the scenic beauty of the nature reserve, consider the cultural heritage of the destination, and improve hospitality facilities, which can enhance tourists’ place attachment and promote the implementation of TERB.

### 7.2. Theoretical Contributions

This study’s theoretical contributions are as follows: (1) Both elements of destination attractiveness significantly influenced the responsible behavior of tourists. The perceived facilities and services, as well as the sightseeing experience, can, to a certain extent, motivate tourists to implement TERB and promote the sustainable development of a destination; (2) place attachment mediates the relationship among the attractiveness of facilities and services, the attractiveness of the sightseeing experience, and TERB. The occurrence of TERB is the result of positive emotions accumulated by tourists after perceiving the attractiveness of the destination. Furthermore, tourists’ intrinsic emotional preferences and psychological dependence can contribute to the formation of TERB; (3) destination attractiveness, as an external stimulus, can affect tourists’ affective attitudes, and when tourists develop high levels of place attachment, they are more inclined to implement TERB.

These findings contribute to the tourism literature by adding new antecedents (destination attractiveness, place attachment, and TERB) and a new causal path for TERB.

### 7.3. Practical Implications

First, the main body governing scenic area management should make efforts to create comfortable and convenient scenic area services using service facilities and its tourism resources which fully demonstrate the resource characteristics of the scenic area.

Second, destinations should focus on the formation of tourists’ place attachment while maintaining their unique attractiveness to effectively promote TERB. Then, destination management organizations should control these factors in a targeted manner to stimulate tourists’ attachment emotions toward the destination, which can promote the practice of TERB when such emotions arise.

Finally, local management should maintain the scenic beauty of the nature reserve, consider the cultural heritage of the destination, and improve hospitality facilities, which can enhance tourists’ place attachment and promote the implementation of TERB.

### 7.4. Conclusions

On one hand, in today’s tourism market, building emotional destination attachment is a key tourism destination branding issue [64]. By improving the current situation of a destination, in areas such as its tourism facilities and accommodation conditions, a destination will improve its attractiveness to tourists, create a good experience for tourists in the tourism process, demonstrate a full understanding of what tourists think and feel, and meet their material and spiritual needs, thereby cultivating tourists’ emotions toward the destination. Relevant departments in shaping the image of tourism destinations should focus on the exposure of special resources and increase their positive image publicity, such that it resonates with tourists, leading to an emotional connection with the destination’s image. For example, the special tourism resources of Wolong National Nature Reserve in Sichuan should be promoted, and the cultural heritage of the nature reserve, such as the dissemination of “panda culture”, should be explored. In this manner, tourists can come to know the unique attraction of the destination and closely connect the destination with their needs.

On the other hand, stimulating tourists to discipline themselves is a more advanced management method. Management should actively perform ecological restoration and protection of nature reserves, promote green tourism, and launch carbon emission reduction volunteer activities. Ecological and environmental awareness cultivation and education activities should also be organized to enhance the sense of social responsibility in tourists and improve their carbon compensation participation behavior.

## 8. Limitations and Future Research Directions

The study has some limitations. The case study site selected in this paper is a national park, which presents limitations in terms of site selection. In addition, this study did not consider differing levels of TERB, as based on individual tourists’ environmental knowledge and other factors. In future studies, the applicability of the findings to other types of tourism destinations can be further verified. Additionally, adding moderating variables, such as environmental knowledge and experience, can be further investigated in future studies.

## Figures and Tables

**Figure 1 behavsci-13-00264-f001:**
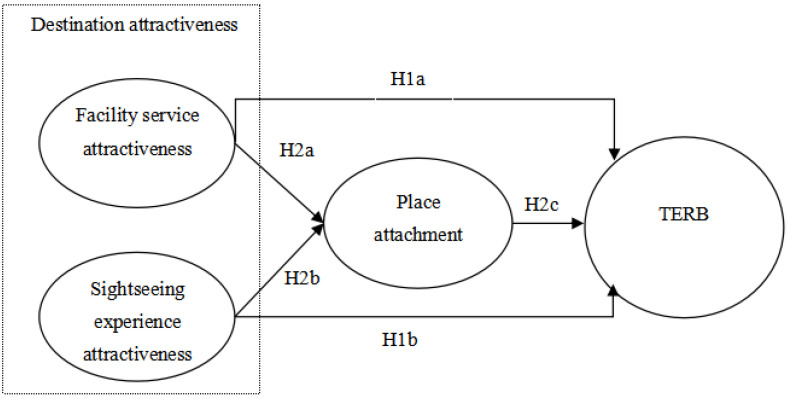
Conceptual Model.

**Figure 2 behavsci-13-00264-f002:**
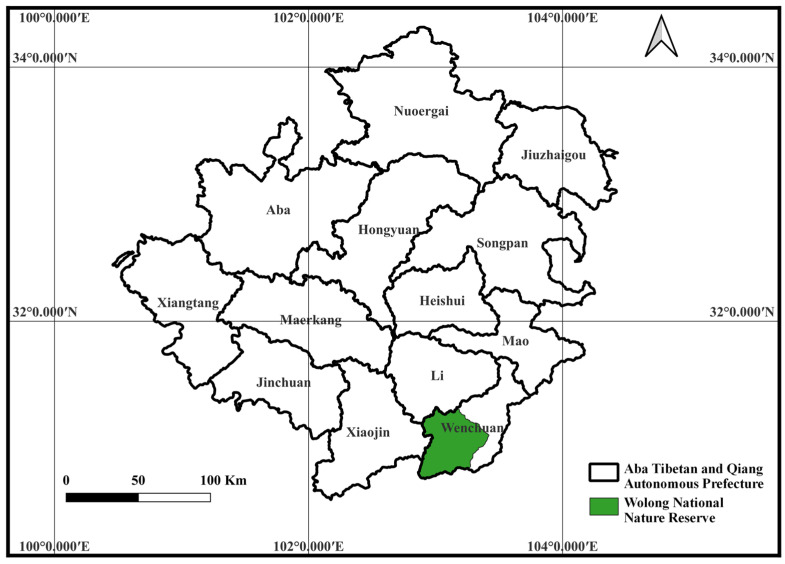
The geographical location of Wolong National Nature Reserve.

**Table 2 behavsci-13-00264-t002:** Fit indices for the structural model.

*X2*/*df*	RMSEA	GFI	AGFI	CFI	IFI	TLI
<3	<0.08	>0.8	>0.8	>0.9	>0.9	>0.9
1.435	0.049	0.956	0.941	0.984	0.987	0.976

**Table 3 behavsci-13-00264-t003:** Results of reliability and validity analysis.

Latent Variables	Measurement Item	Standardized Load	*t*-Value	Composite Reliability	Cronbach’s α Value	AVE
Facility service attractiveness	A1 A2 A3	0.728 0.765 0.828	8.185 7.764	0.74	0.815	0.51


Sightseeing experience attractiveness	B1 B2	0.688 0.755	- 8.357	0.69	0.746	0.50

Place attachment	C1	0.728		0.87	0.887	0.63
	C2	0.813	11.256			
	C3	0.826	11.574			
	C4	0.792	10.689			
TERB				0.85	0.814	0.62
	E1	0.774	10.768			
	E2	0.768	11.462			
	E3	0.845	12.316			
	E4	0.802				
Overall scale					0.916	

**Table 4 behavsci-13-00264-t004:** Results of the validation of the hypotheses of the model.

Path Relationship	Standardized Path Coefficient	S.E.	C.R.	*p*
H1a: Facility service attractiveness → Place attachment	0.835	0.146	3.187	***
H1b: Sightseeing experience attractiveness → Place attachment	0.792	0.165	3.731	***
H2a: Facility service attractiveness → TERB	0.699	0.476	4.586	***
H2b: Sightseeing experience attractiveness → TERB	0.851	0.254	3.496	***
H2c: Place attachment → TERB	0.822	0.153	4.674	***

Note: *** indicates *p* < 0.001.

**Table 5 behavsci-13-00264-t005:** Test of the mediating effect.

Path	Indirect Effect	Percentile Inspection of 95%
Facility service attraction → Place attachment → TERB	0.072	[0.064, 0.116]
Sightseeing experience attractiveness → Place attachment → TERB	0.081	[0.053, 0.097]

## Data Availability

The data used to support the findings of this study are available from the corresponding author upon request.

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
