# Peer review of "An Integrated Model of Destination Attractiveness and Tourists’ Environmentally Responsible Behavior: The Mediating Effect of Place Attachment"

_behavsci, 2023, doi:10.3390/bs13030264_

Round 1

Reviewer 1 Report

The article discusses TERB from the perspective of destination attractiveness. The topic sounds interesting, but I suggest authors should think seriously about the following suggestions:

1.     The Introduction part lacks logical integration. Elaboration on the significance of destination attractiveness, the reason to apply self-regulatory attitude theory as well as objectives of this research should be clearly presented.

2.     As to 2.1 Relationship between destination attractiveness and TERB, a number of attributes concerning destination attractiveness are listed, but why do you finally choose the two dimensions? You need to explain.

3.     3.1 Theoretical foundation should be moved to 2. Literature review and hypothesis, for all the hypothesis were developed based on the theoretical framework. The theory should also be mentioned in the Introduction part.

4.     In the 3.2 Scale design section, it is better to present a detailed measurement table with all items listed on it.

5.     The biggest problems lie in 3.3 Data collection. First, you need to verify the representativeness of the survey site, i.e., Wolong National Nature Reserve. “It has a large human population and a certain guarantee of sample diversity”. How large and what is certain? Second, the valid questionnaire number is not enough (i.e. 226), and the effective rate is not satisfying, too. Third, the proportion concerning the educational background of respondents is not so balanced. Maybe the authors need to perform another new data procedure in the next revision.

6.     It’s pity that normality test and CMB/CMV test are omitted. Please add it.

7.     The AVE values of facility service attractiveness and sightseeing experience attractiveness are lower than 0.5. These results did not meet the relevant requirement.  Why?

8.     As per 7. Conclusion and discussion, the authors mixed findings, conclusions, and practical implications. Besides, theoretical contributions as an essential part is missing. It’s strongly suggested that these parts should be illuminated respectively with subchapters.

9.     As for the reference, regarding the limited number, more references could be included. For instance,

Zheng, W.; Qiu, H.; Morrison, A.M.; Wei, W.; Zhang, X. Landscape and unique fascination: A dual-case study on the antecedents of tourist pro-environmental behavioral intentions. Land 2022, 11, 479.

This reference provides explicit description on destination attractiveness, which may enhance theoretical argument in this paper.

Additionally, there are lots of TERB articles in 2022, authors could read them to update this paper’s reference current status. Through the dialogue of the recent TERB literature, the novelty of this paper can be better presented. In this sense, it’s a nice suggestion.

Author Response

Response to Reviewer 1 Comments

Point 1: The Introduction part lacks logical integration. Elaboration on the significance of
destination attractiveness, the reason to apply self-regulatory attitude theory as well as objectives of this research should be clearly presented.

Response 1: Thank you for the above suggestion. In the Introduction part, considering the
reviewers suggestion, we have added an elaboration on the significance of destination
attractiveness on P2, Line 50-55, the reason to apply self-regulatory attitude theory on P3, Line 99-105, and clearly present objectives of this research on P3, Line 112-117, to better improve logical integration.

Point 2: As to 2.1 Relationship between destination attractiveness and TERB, a number of attributes concerning destination attractiveness are listed, but why do you finally choose the two dimensions? You need to explain.

Response 2: Thank you for your good questions. In the literature review of destination attraction, the dimensions of destination attraction measured by previous scholars are added on P4, Line 154-170, and the commonly used two dimensions are selected by referring to the research of existing scholars.

Point 3: Theoretical foundation should be moved to 2. Literature review and hypothesis, for all the hypothesis were developed based on the theoretical framework. The theory should also be mentioned in the Introduction part.

Response 3: Thank you for the above suggestion. We have moved theoretical foundation to 2. Literature review and hypothesis on P3, Line 118-133, and the theory is mentioned in the Introduction part on P3, Line 99-105.

Point 4: In the 3.2 Scale design section, it is better to present a detailed measurement table with all items listed on it.

Response 4: Thank you for your excellent suggestion. We have added a detailed measurement table in the 3.2 Scale design section with all items listed on it on P8, Line 291. (see Table 1. Measurement elements.)

Point 5: The biggest problems lie in 3.3 Data collection. First, you need to verify the
representativeness of the survey site, i.e., Wolong National Nature Reserve. It has a large human population and a certain guarantee of sample diversity. How large and what is certain? Second, the valid questionnaire number is not enough (i.e. 226), and the effective rate is not satisfying, too. Third, the proportion concerning the educational background of respondents is not so balanced. Maybe the authors need to perform another new data procedure in the next revision.

Response 5 : Thank you for your good questions. We have added a section of 3.1 Study Area section to make a supplementary explanation to the case on P6, Line 248-265. And we also make a supplementary investigation,collected enough valid questionnaires,and re-analyzed the data on P8, Line 290-294.

Point 6: Its pity that normality test and CMB/CMV test are omitted. Please add it.

Response 6Thank you for your good suggestions. We have performed normality test and CMV test, and the test results have been briefly explained in this paper.

Point 7: The AVE values of facility service attractiveness and sightseeing experience attractiveness are lower than 0.5. These results did not meet the relevant requirement. Why?

Response 7Thank you for your questions. We have added the AVE values of two dimensions both reached 0.5 after the number of valid questionnaires on P10, Line 343-350..

Point 8: As per 7. Conclusion and discussion, the authors mixed findings, conclusions, and practical implications. Besides, theoretical contributions as an essential part is missing. It s strongly suggested that these parts should be illuminated respectively with subchapters.

Response 8Thank you for your suggestions. We have divided chapter 7 conclusion and discussion into four subchapters and theoretical contributions part is added .

Point 9: As for the reference, regarding the limited number, more references could be included. For instance, Zheng, W.; Qiu, H.; Morrison, A.M.; Wei, W.; Zhang, X. Landscape and unique fascination: A dual-case study on the antecedents of tourist pro-environmental behavioral intentions. Land 2022, 11, 479.This reference provides explicit description on destination attractiveness, which may enhance theoretical argument in this paper. Additionally, there are lots of TERB articles in 2022, authors could read them to update this papers reference current status. Through the dialogue of the recent TERB literature, the novelty of this paper can be better presented. In this sense, its a nice suggestion.

Response 9Thank you for your good suggestions. We have added more references to update this papers reference current status.

Reviewer 2 Report

See the details in the attachment!

Author Response

Response to Reviewer 2 Comments

Point 1:  The research background is described sufficiently. Need more rationale by presenting the facts why tourists’ environmentally responsibility behaviour is needed to be research in the selected research location! Please elaborate more deeply on the phenomena studied using the indicators!

Response 1: Thank you for the above suggestion. Considering the Reviewer’s suggestion, we made a supplementary explanation and provide more rationale in the 3.1 Study Area section and elaborate more deeply on the phenomena studied using the indicators on P6, Line247-265.

Point 2:  Need more relevant references on the variables used in the research.

Response 2: Thank you for your suggestions. We have added more relevant references on the variables used in the research. And we also have added a detailed measurement table in the 3.2 Scale design section with relevant references listed on it on P8, Line291 (see Table 1. Measurement elements.)

Point 3 :Method used needs to be more specific; and Qualitative method must be appropriate: (Survey Research, Correlational Research, Quasi-Experimental Research / Comparative Studies, or Experimental Research);Specify the research design used and explain the reasons of using this research approach as well as how the data analysed! Hypothesis must be specifically drawn up based on the research purposes and model adopted Sampling technique is not clear. How did you decide the number 309 and 226? Use Slovin to get the right number and confirm the validity and realibility.

Response 3: We appreciate it very much for this good suggestion, and we have done it according to your ideas. First, we used quantitative research methods. Second, we have supplemented the process of questionnaire design and questionnaire collection on P8, Line294-305. Next, convenience sampling was conducted at the case site, and we supplemented the process of questionnaire processing. We perform confirmatory factor analysis by using maximum likelihood estimation to separately measure the convergent and discriminant validity of the scale.

Point 4: The results need to be specified as per research objectives / hypothesis.

Response 4: Thank you for your suggestions. We have divided chapter 7 conclusion and discussion into four subchapters theoretical contributions part is added on P12,Line412-426 and results are specified as per research objectives / hypothesis.   .  

Point 5: Need to add references research methods used especially the sampling technique and research instruments.

Response 5: Thank you for your suggestions. We have added more references about the sampling technique and research instruments.

Point 6: Conclusion MUST discuss the following areas:

  1. Researcher's view why the case is interesting to be investigated
  2. Write conclusions based on the research questions and reflect to the literature reviews adapted in this research
  3. Write research Implications and limitations
  4. Add suggestion for further scientific researches related to this finding

Response 6: We appreciate it very much for this good suggestion, and we have done it according to your ideas. First, in the 7.1.discussion section, we add a supplementary explanation to the importance of TERB research on P11,Line395-398. We also add lots of TERB articles in 2022, authors could read them to update this paper’s reference current status. Through the dialogue of the recent TERB literature, the novelty of this paper can be better presented. Second, we have divided chapter 7 conclusion and discussion into four subchapters. Theoretical contributions and research implications are also added on P12, Line412-426. At last, we add limitations and suggestion for further scientific researches related to this finding.

Reviewer 3 Report

Based on my point of view, this paper fulfill the requirement needed for this journal. Statistics techniques used in this study is adequate but you need to mention the type of sampling you are using. Another thing, it will be much better if there is a map included to show the location of the study area.

Author Response

Response to Reviewer 3 Comments

Point 1: Based on my point of view, this paper fulfill the requirement needed for this journal. Statistics techniques used in this study is adequate but you need to mention the type of sampling you are using. Another thing, it will be much better if there is a map included to show the location of the study area.

Response 1: Thank you for your suggestions. We conducted convenience sampling at the case site and mention. Considering the Reviewer’s suggestion, We made a supplementary map by ArcGIS to show the location of the study area on P7,Line266.(Data Sources:The vector boundary data of the administrative division of Aba Tibetan and Qiang Autonomous Prefecture and Wolong National Nature Reserve are from the National Basic Geographic Information Center and the specimen resource sharing platform of China National Nature Reserve, respectively)
